# STAT-3 is necessary for IL-27-mediated macrophage suppression but does not represent a therapeutic target for *E. coli*-induced neonatal sepsis

Madhavi Annamanedi,[1] Jessica M. Povroznik,[1] Cory M. Robinson[1,2]

**ABSTRACT**    Interleukin (IL)-27 is a heterodimeric immunoregulatory cytokine expressed at elevated levels early in life that compromises bacterial clearance and promotes severe outcomes during neonatal sepsis. In turn, IL-27Rα-deficient neonatal mice exhibit better control of bacteria, reduced systemic inflammation, and improved outcomes. IL-27 primarily activates and signals through either Signal transducer and activator of transcription (STAT)-1 or STAT-3 in macrophages. Targeted deletion of STAT-3 in macrophages has been reported to improve responsiveness to Lipopolysaccharide (LPS) and promote Th1 activity. As such, in the present study, we investigated the role of STAT-3 signaling in IL-27-mediated suppression of bacterial clearance and lysosomal activity in neonatal macrophages during *in vitro* infection and sepsis. Bone marrow-derived macrophages from myeloid-specific STAT-3 deletion in neonatal mice (LysMcre-STAT-3$^{fl/fl}$) showed improved control of intracellular bacteria and lysosomal acidification. Consistent with these findings, *E. coli*-infected neonatal LysMcreSTAT-3$^{fl/fl}$ mice demonstrated improved bacterial clearance, but conversely, increased inflammatory response and mortality compared with neonatal mice with intact STAT-3 signaling. Pharmacological inhibition of STAT-3 in WT neonatal mice using S32-201 resulted in the inability to clear bacteria in either the blood or spleen relative to control mice. This study revealed that STAT-3 is necessary for IL-27 suppression of macrophage-mediated bacterial killing, but neither myeloid-specific nor global STAT-3 inhibition during neonatal sepsis achieves the same outcome as loss of IL-27 signaling. This suggests that STAT-3 is not a promising therapeutic target to mitigate IL-27 activity in early life infection.

**IMPORTANCE** The neonatal period is a time in which newborns have increased vulnerability and the highest risk of death from infection. This includes sepsis for which there is a considerable global burden of disease. We have determined that the cytokine interleukin (IL)-27 is expressed at elevated levels in the first days of life and continues to rise during experimental bacterial neonatal sepsis. Neonatal mice that cannot respond to IL-27 exhibit improved outcomes. In this work, we have investigated the influence of STAT-3 on control of bacteria and inflammation during IL-27 signaling in neonates. It is critical that we understand mechanisms that underlie neonatal susceptibility to infection so that we can identify new targets for therapeutic intervention. Here, we define the value of STAT-3 in approaches to targeted therapies for bacterial neonatal sepsis.

**KEYWORDS**    interleukin-27, neonatal sepsis, STAT-3, lysosome, bacteria, inflammation, blood, spleen

S epsis is the leading cause of death in hospitalized patients and remains a major cause of morbidity and mortality among neonates (1). The neonatal period is a vulnerable time in which newborns have increased susceptibility and the highest risk of

**Peer Reviewer** Modher Nagem Abed Abed, Baghdad university, Baghdad, Iraq

Address correspondence to Cory M. Robinson, cory.robinson1@hsc.wvu.edu.

The authors declare no conflict of interest.

See the funding table on p. 11.

death from infection. This is consistent with an immune system that functions uniquely different from older children and adults. While differences in cytokine production in response to microbial stimuli, such as reduced capacity of antigen-presenting cells to stimulate T cells, and increased levels of anti-inflammatory cytokines reflective of more Th2 polarizing activity, highlight some aspects of the current understanding (2–4), the reasons that underlie early life susceptibility to infection remain incompletely defined. Multiple studies have identified the value of IL-27 as a biomarker and predictor of pediatric infection and neonatal sepsis specifically (5–8). Interleukin (IL)–27 belongs to the IL-12 cytokine family and is a heterodimeric cytokine composed of IL-27p28 and Epstein–Barr virus-induced gene 3 (EBI3), which signals through a receptor complex of gp130 and IL-27 receptor (IL-27R) -α (9). Although it was first identified for a proinflammatory role in promoting differentiation of Th1 cells and production of IFN-γ, IL-27 also has suppressive activity toward both innate and adaptive immune cells (10). This suppressive role has been documented in animal models of infectious and chronic diseases (11–13).

We have established that IL-27 expression by human and murine macrophages is elevated early in life (14, 15). Similarly, mice have increased levels of IL-27 in the serum during the neonatal phase, and these levels continue to rise during infection (14). In both neonatal and adult murine sepsis studies, IL-27 levels have been found to increase and are consistent with compromised bacterial clearance, elevated inflammation, and a lack of protective immunity (14, 16, 17). IL-27Rα-deficient neonatal pups specifically exhibit superior bacterial clearance in the blood and peripheral tissues, a reduced inflammatory response, and improved weight gain and survival compared with WT pups with intact IL-27 signaling (14, 18). A transcriptional profiling of the spleen determined that a macrophage-enriched population of myeloid cells contributes significantly to the inflammatory response that is minimized in the absence of IL-27 signaling during infection (18). However, the exact mechanisms that regulate compromised bacterial clearance and heightened inflammatory response in the presence of elevated levels of IL-27 are unclear.

Upon binding the receptor on target cells, IL-27 signals through signal transducer and activator of transcription (STAT) proteins. Depending on the immune cell type, IL-27 primarily stimulates phosphorylation of STAT-1 and STAT-3. IL-27-induced inflammatory cytokine expression is STAT-1-dependent in human monocytes (19–21). In naive T helper (Th) cells, IL-27 activates both STAT-1 and STAT-3, but in activated Th cells, it induces only STAT-3 (10). In an earlier study, we demonstrated that in human neonatal macrophages, IL-27 activates both STAT-1 and STAT-3, yet IL-27-mediated induction of the T-cell suppressive gene indoleamine dioxygenase (IDO) required STAT-3 (22). In other works, targeted deletion of STAT-3 in macrophages led to increased responsiveness to LPS and enhanced Th1 activity (23, 24). The anti-inflammatory effects of IL-10, another suppressive cytokine, are also mediated by STAT-3 (24, 25). We therefore hypothesized that compromised control of bacteria in response to IL-27 during neonatal sepsis requires STAT-3 signaling, and neonatal mice lacking the ability to activate STAT-3 would mirror the sepsis-resistant phenotype observed in IL-27Rα-deficient neonatal pups.

## MATERIALS AND METHODS

### Mice

C57BL/6 (WT), IL-27Rα-deficient (KO), STAT-3$^{fl/fl}$ homozygous for the loxP-flanked (floxed) *STAT-3* gene, and LysM-Cre$^{+/+}$ mice carrying a *Cre* transgene under the control of the *LysM* promoter on a C57BL/6 genetic background were purchased from Jackson Laboratory (Bar Harbor, ME, USA) and maintained under specific pathogen-free conditions in the vivarium at West Virginia University Health Sciences Center. Mice were maintained on a 12 h light/dark cycle and were fed/watered *ad libitum*. Conditional STAT-3 deficient mice were generated by crossbreeding STAT-3$^{fl/fl}$ and LysM-Cre$^{+/+}$ mice. Pups born were further genotyped using gene-specific primers (Jackson Laboratory, *STAT*-3 FP: GGGGTGAGAGTT

ACCGTGAA; RP: CACACACACACAAGCCATCA and LysMCre FP: CTTGGGGCTGCCAGAATTTCT C; RP: CCCAGAAATGCCAGATTACG) and screened for *lysMcreSTAT-3*<sup>fl/fl</sup> genotype. Both male and female 4-day-old pups were used for experimental infection. All procedures were approved by the West Virginia University Institutional Animal Care and Use Committees and conducted in accordance with the recommendations from the *Guide for the Care and Use of Laboratory Animals* by the National Research Council (26).

## Generation of mouse bone marrow-derived macrophages

Neonatal bone marrow isolation was performed as described previously (27). Briefly, neonatal bone marrow cells were seeded and differentiated into bone marrow-derived macrophages (BMDMs) using L929 supernatant conditioned Dulbecco's modified eagle medium (DMEM). On day 5, BMDMs were detached with 0.05% trypsin-EDTA and used as described.

## Immunofluorescent staining and microscopy

Cells were fixed in 2% paraformaldehyde solution for 30 min at room temperature. They were then washed twice in PBS followed by a 30 min incubation in 1% bovine serum albumin solution (Sigma-Aldrich, St. Louis, MO, USA). Cells were permeabilized with 0.1% Triton X-100 and labeled with mouse STAT-3 #9139 (Cell Signaling Technology, Danvers, MA, USA) and mouse phospho-STAT-3 antibody #9145 (Cell Signaling Technology, Danvers, MA, USA). Excess, unbound antibody solution was washed away with PBS, and the cells were incubated for 45 min at room temperature with secondary anti-mouse IgG Alexa Fluor 568 (Ex: 579 nm, Em: 603 nm) and anti-rabbit IgG Alexafluor 488 (Ex: 499 nm, Em: 520 nm) (ThermoFisher Scientific, Waltham, MA, USA). Nuclear staining was performed with 4',6-diamidino-2-phenylindole (DAPI). Finally, the samples were washed to remove excess nuclear staining solution and mounted on microscope slides for imaging. The labeled cells were visualized using the Zeiss 710 confocal microscope.

## *In vitro* bacterial clearance assay

Bacterial clearance by BMDMs *in vitro* was measured as described previously (14). WT and LysMcreSTAT-3<sup>fl/fl</sup> BMDMs were seeded in a 96-well plate at a density of $5 \times 10^4$ cells per well. BMDMs were infected with *Escherichia coli* O1:K1:H7 (ATCC Manassis, VA, USA) engineered to express luciferase and described previously (14), at a multiplicity of infection (MOI) of 100 for 1 h at 37°C and 5% $CO_2$. The medium was then replaced with fresh medium supplemented with gentamicin (100 µg/mL), and the cultures were returned to incubation for an additional 5 h. Bacterial luminescence was measured using a Molecular Devices SpectraMax iD3 (San Jose, CA, USA) at 2 and 6 h post-infection. Bacterial clearance was estimated by subtracting luminescence values at 6 h post-infection (1 h after addition of gentamicin) from 2 h post-infection measurements; an increase in the delta is directly proportional to bacterial killing.

## Labeling and identification of lysosomes

WT, IL-27Rα KO, and LysMcreSTAT-3<sup>fl/fl</sup> BMDMs were seeded in a 35 mm quad dish (iBidi, Gräfelfing, Germany) at a density of $2 \times 10^5$ cells per quadrant in a volume of 500 µL of complete DMEM without phenol red or antibiotics. The bacterial inoculum was prepared at the MOI of 25 using a pre-titered stock of *E. coli* O1:K1:H7 stored at −80°C. Bacteria were washed twice in 1 mL of PBS by centrifuging at 2,000×*g* for 5 min. The bacterial pellet was resuspended in 50 µL of PBS and labeled with the pH-sensitive dye pHrodo green (Thermo Fisher Scientific, USA) to a final concentration of 500 µM. The bacterial cells were incubated for 20 min in the dark and then washed four times with 1 mL of PBS by centrifugation. The bacterial pellet was resuspended in 500 µL of complete DMEM without phenol red and added to the BMDMs. After 5 h of incubation at 37°C with 5% $CO_2$, 200 ng of LysoTracker Red (ThermoFisher Scientific) was added to the culture.

Phagocytosed bacteria and acidified lysosomes were visualized by Zeiss 710 confocal fluorescence microscopy.

## Neonatal murine sepsis infection model

Neonatal STAT-3$^{fl/fl}$ and LysMcreSTAT-3$^{fl/fl}$ pups at the ages of 4 days were infected subcutaneously in the scapular region with *E. coli* strain O1:K1:H7 using a 28-gauge insulin needle as described previously (14). The bacteria were washed with PBS, centrifuged at 2,000×*g* for 5 min, and suspended in a volume of PBS equivalent to an inoculum of 50 µL/mouse. The target inoculum was $10^5$ CFUs per mouse, and actual inoculums, as determined by standard plate counts, ranged from 1 to 3 × $10^5$ CFUs per mouse pup. Control mice received vehicle (PBS). For inhibitor studies, pups were pretreated with STAT-3 inhibitor, S32-201 (Sigma-Aldrich, USA) at 5 mg/kg for 2 h prior to infection.

The weights of mice were recorded immediately prior to infection and then again at 24 h post-infection prior to euthanasia as indicated in the figure legend. Blood glucose was measured using an AlphaTrack3 blood glucose monitoring system (Zoetis, MI, USA), and the remaining blood was deposited in microcentrifuge tubes that contained 5 µL of 500 mM ethylenediamine tetraacetate acid (EDTA) and placed on ice for measurement of bacterial burdens and collection of serum. Spleens isolated from pups were placed in PBS on ice and further homogenized with a handheld pestle motor (Kimble Chase, NJ, USA). The bacterial burden in the blood and spleens was enumerated by serial dilution and standard plating on TSA plates. In some experiments, spleens were collected in TriReagent (Sigma-Aldrich, USA) and stored at −80°C for gene expression analysis. For survival experiments, pups were monitored for 24 h post-infection.

## RNA isolation and gene expression analysis

Spleens were thawed and homogenized in TriReagent. RNA was isolated using the commercial product protocol. Briefly, the upper aqueous layer following phase separation was mixed with an equal volume of 75% ethanol and transferred to E.Z.N.A. RNA isolation columns (Omega Biotek, Norcross, GA, USA). The manufacturer's instructions were followed to complete tissue RNA isolation. iScript cDNA synthesis reagents (Bio-Rad, Hercules, CA, USA) were used to generate first strand cDNA according to the manufacturer's protocol. Real-time cycling of reactions that included cDNA from the above preparation diluted 1:3 in nuclease-free water, gene-specific primer probe sets (Applied Biosystems, Foster City, CA, USA), and iQ Supermix (Bio-Rad) was performed in triplicate using a StepOnePlus (Applied Biosystems, Foster City, CA, USA) real-time detection system. Gene-specific amplification was normalized to that of *actB* as an internal reference gene and expressed as log$_2$ relative gene expression compared to control spleens using the formula $2^{-\Delta\Delta Ct}$.

## Cytokine detection by MSD

IL-6 and IL-10 serum levels were measured using multiplexed electrochemiluminescence U-plex reagents according to the manufacturer's protocol (MesoScale Discovery [MSD], Rockville, MD, USA). Results were analyzed using MSD Discovery Workbench software (v4.0.13). Protein standards were assayed in parallel with samples.

## Statistical analysis

All data sets were analyzed with the appropriate parametric or nonparametric test as indicated in the figure legend using Prism 8 (GraphPad, San Diego, CA). The threshold for statistical significance was set at alpha = 0.05.

## RESULTS

### STAT-3 activation is inhibited in IL-27-stimulated BMDMs from LysMcre-STAT-3$^{fl/fl}$ neonatal mice

We first evaluated the responsiveness of STAT-3 signaling to IL-27 in neonatal BMDMs and analyzed the level of inhibition in cells isolated from LysMcreSTAT-3$^{fl/fl}$ mouse pups. STAT-3 becomes activated by phosphorylation at Tyr705 upon interaction with a variety of cytokines, including IL-27 (21, 28, 29). Under normal conditions, phosphorylation of STAT-3 promotes dimerization and translocation to the nucleus to regulate the transcription of responsive genes. STAT-3$^{fl/fl}$ mice possess *loxP* sites that flank exons 18–20 that include Tyr705 and the SH2 domain required for nuclear translocation (30). When these mice are bred with LysMcre mice that express Cre recombinase under control of the lysozyme 2 gene (*lyz2*), the *loxP*-flanked exons of STAT-3 are deleted. Expression of the *lyz2* gene is myeloid-restricted (31). BMDMs isolated from STAT-3$^{fl/fl}$ mice in the absence of Cre recombinase that have a WT phenotype demonstrated a baseline level of nuclear phosphorylated STAT-3 that was markedly increased upon stimulation with IL-27 (Fig. 1A). In contrast, BMDMs from LysMcreSTAT-3$^{fl/fl}$ neonatal mice showed the absence of STAT-3 phosphorylation and nuclear translocation even after stimulation with IL-27 (Fig. 1B). This immunofluorescence staining demonstrated that in neonatal BMDMs, IL-27 activates STAT-3 and further confirmed that LysMcreSTAT-3$^{fl/fl}$ mice are unable to activate STAT-3 in an IL-27-responsive manner.

### The absence of STAT-3 activation promotes bacterial killing and lysosomal acidification in neonatal BMDM

We have previously shown that neonatal BMDMs from IL-27Rα$^{-/-}$ mice eliminated *E. coli* with increased efficiency during infection compared with those from WT (14). Here, we examined *in vitro* bacterial clearance in neonatal LysMcreSTAT-3$^{fl/fl}$ BMDMs using the same established assay and luciferase-expressing *E. coli*. We found that BMDMs

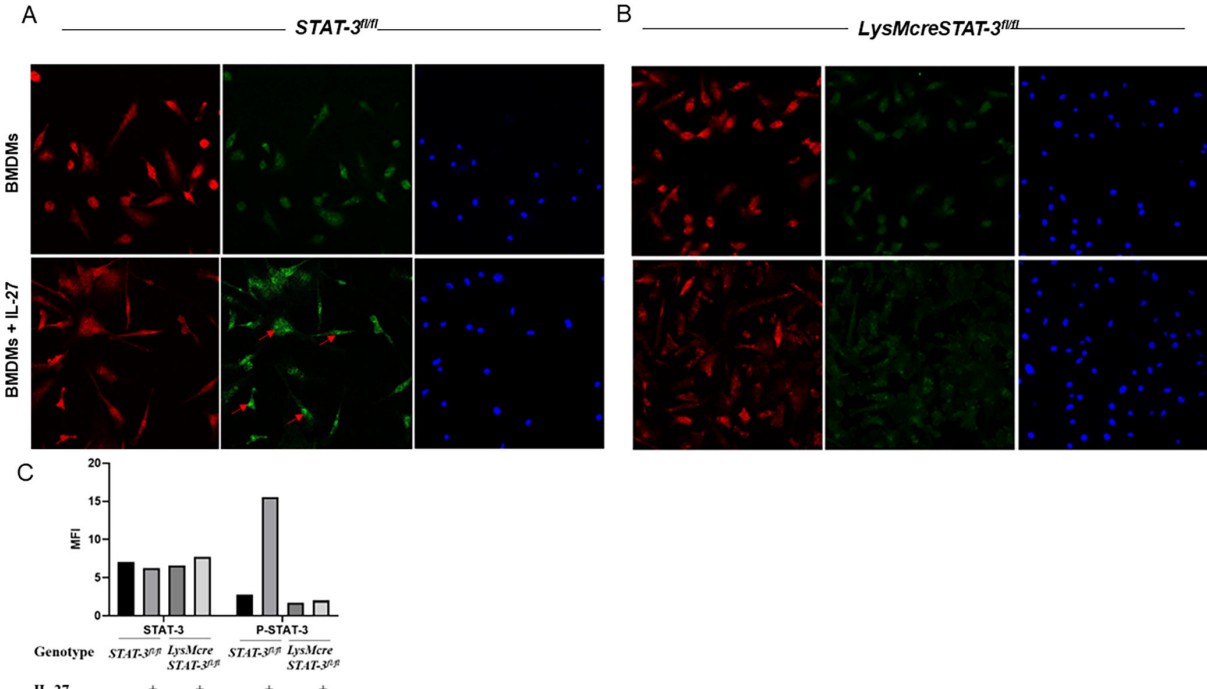

FIG 1 Inhibition of STAT-3 activation in IL-27-stimulated BMDMs from LysMcreSTAT-3$^{fl/fl}$ neonatal mice. BMDMs were treated with 100 ng/mL IL-27 cytokine for 4 h. Representative 40× confocal microscopy images of immunofluorescence staining for total STAT3 (red), P-STAT3 (green), and DAPI staining of nuclei (blue) in the BMDMs from (A) Stat-3$^{fl/fl}$ and (B) LysMcreSTAT-3$^{fl/fl}$ neonatal mice. (C) Fluorescence quantification of STAT-3 and P-STAT-3 signal in BMDMs from Stat-3$^{fl/fl}$ and LysMcreSTAT-3$^{fl/fl}$ from panels A and B. Red arrows indicate phospho-STAT3 translocated into the nucleus of IL-27-treated neonatal BMDMs. Scale bar: 10 µm.

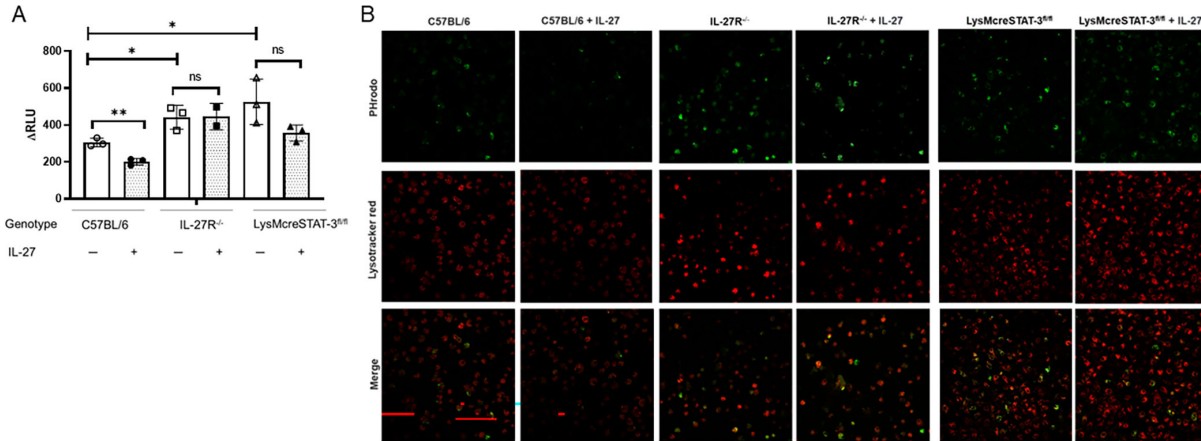

**FIG 2** BMDMs from LysMcreSTAT-3^fl/fl exhibit increased lysosomal acidification and bacterial clearance. Macrophages were derived from bone marrow progenitors obtained from WT, LysMcreSTAT-3^fl/fl, or IL-27Rα^-/- neonatal mice. (A) Cells were seeded in 96-well plates, treated for 2 h with or without IL-27 (100 ng/mL) and infected with luciferase-expressing *E. coli* O1:K1:H7 at a multiplicity of infection (MOI) of 100. At 1 h post-infection, the medium was replaced with fresh medium that contained gentamicin (100 µg/mL). A change in relative luminescent units (ΔRLU) was measured by deducting luminescent values at 6 h (bacteria remaining) from 2 h post-infection (starting bacterial load). Statistical significance in the 95% confidence interval (CI) was determined using individual unpaired t tests; exact *P*-values are shown. (B) For the lysosomal acidification assay, BMDMs were infected with pHrodo-labeled *E. coli* O1:K1:H7 at a multiplicity of infection (MOI) of 25. At 5 h post-infection, LysoTracker Red (200 ng/mL) was added to the culture. Representative 20× confocal images of neonatal BMDMs from C57BL/6, IL-27Rα^-/-, and LysMcreSTAT-3^fl/fl pups showing bacteria in acidified compartments (green), acidified lysosomes (red), or the merged combination.

from LysMcreSTAT-3^fl/fl neonatal mice that lack STAT-3 activation and nuclear translocation exhibited more bacterial killing capability when compared with BMDMs from WT neonatal mice (Fig. 2A). Increased bacterial killing was consistent with improved localization of bacteria to lysosomes and lysosomal acidification (Fig. 2B; Fig. S1). BMDMs from LysMcreSTAT-3^fl/fl neonatal mice showed an increased abundance of pHrodo-labeled bacteria that only fluoresce in acidified compartments, as well as enhanced lysosomal acidification compared with WT neonatal BMDMs (Fig. 2B; Fig. S1). Because of the genetic variability among WT, IL-27Rα^−/−, and LysMcreSTAT-3^fl/fl genotypes, phagocytic ability and lysosomal acidification are also expected to be dissimilar prior to the IL-27 cytokine stimulation (Fig. 2B). These findings from LysMcreSTAT-3^fl/fl macrophages were comparable to those observed in BMDMs from IL-27Rα^−/− pups as a historical positive control (Fig. 2B). The addition of IL-27 cytokine significantly reduced the killing of bacteria (Fig. 2A and B) and lysosomal acidification in the WT BMDMs but not in the IL-27Rα^−/− or lysMcreSTAT-3^fl/fl macrophages (Fig. 2B). Collectively, these findings demonstrate that STAT-3 activation and IL-27 have comparable regulatory influences on bacterial trafficking to lysosomes and lysosomal activity with consequences to bacterial killing and suggest that STAT-3 is the common signaling intermediate.

## Improved bacterial clearance with enhanced morbidity was observed in septic neonatal mice that lack myeloid-specific STAT-3 activation

The influence of STAT-3 on macrophage control of bacteria suggested that it may represent a target for therapeutic intervention during neonatal sepsis. To evaluate the impact of STAT-3 on *in vivo* bacterial clearance and sepsis outcomes, LysMcreSTAT-3^fl/fl neonates were infected with *E. coli* and compared with vehicle-treated and infected STAT-3^fl/fl (WT) neonates. Both infected genotypes lost weight similarly compared with uninfected controls (Fig. 3A). In our neonatal sepsis model, infected neonatal pups become hypoglycemic like septic human babies (32). Vehicle-treated neonates maintained blood glucose levels, although significant hypoglycemia was observed in both infected STAT-3^fl/fl and LysMcreSTAT-3^fl/fl neonates (Fig. 3B). There was a 68% and more significant 99.5% reduction in the bacterial burden observed in the blood and spleens of infected LysMcreSTAT-3^fl/fl neonates, respectively, when compared with

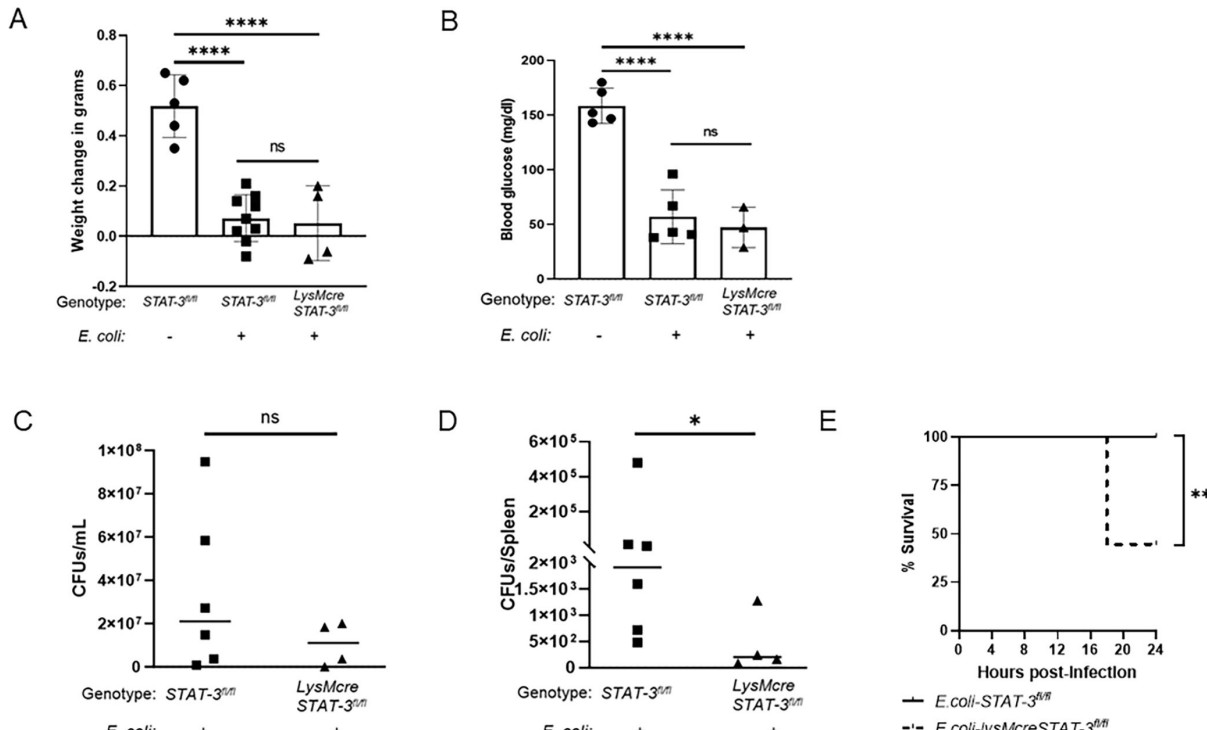

**FIG 3** Inhibition of STAT-3 in myeloid cells promotes bacterial clearance but results in lethality during neonatal sepsis. Stat-3$^{fl/fl}$ and LysMcreSTAT-3$^{fl/fl}$ ($n = 3$ in each treatment group) pups were infected subcutaneously with *E. coli* O1:K1:H7 (1–3 × 10$^5$ CFUs/mouse) or vehicle control on day 4 of life. Weight change (A), blood glucose levels (B), bacterial burdens in the blood (C) or spleen (D), and survival rate (E) were measured through 24 h of infection. The data from three combined experiments performed separately are shown. Statistical significance in the 95% CI was determined using a Mann–Whitney test; exact *P*-values are shown.

infected STAT-3$^{fl/fl}$ neonates (Fig. 3C and D). In the blood, STAT-3$^{fl/fl}$ neonates have 3.33 × 10$^7$ mean CFU/mL, whereas LysMcreSTAT-3$^{fl/fl}$ neonates have 1.05 × 10$^7$ mean CFU/mL. Spleens of STAT-3$^{fl/fl}$ and LysMcreSTAT-3$^{fl/fl}$ neonates showed 8.28 × 10$^4$ and 4.40 × 10$^2$ mean CFU/tissue, respectively. Despite the improvement in control of systemic bacteria that was consistent with *in vitro* findings, LysMcreSTAT-3$^{fl/fl}$ neonates surprisingly had a 55.5% mortality rate within 24 h of infection (Fig. 3E). In contrast, all the infected STAT-3$^{fl/fl}$ neonates survived through 24 h (Fig. 3E). These findings demonstrate that although improved bacterial clearance was similar to that observed with IL-27Rα$^{-/-}$ neonatal mice (14), morbidity and mortality were not similarly improved with disruption of STAT-3 activation. In fact, increased mortality is observed with neonatal mice that lack STAT-3 activation in myeloid cells during sepsis. We considered the possibility that the inflammatory response was uncontrolled and pathological. Thus, we analyzed cytokine expression in the spleen and circulating levels in the serum. Gene expression of IL-1β and IL-6 increased more significantly in the spleen during infection in LysMcreSTAT-3$^{fl/fl}$ mice (Fig. 4A). This is consistent with increased bacterial clearance in the spleen but may also contribute to local tissue damage. IL-10 expression levels increased in infected pups likely as a response to the inflammatory environment, but overall levels were comparable for both genotypes. In the periphery, serum levels of IL-6 and IL-10 were significantly increased during infection, but at comparable levels in both STAT-3$^{fl/fl}$ and LysMcreSTAT-3$^{fl/fl}$ neonates (Fig. 4B and C). However, in contrast, IL-27Rα$^{-/-}$ mice exhibit reduced levels of inflammatory cytokines during infection, consistent with our prior report (Fig. 4B and C) (14).

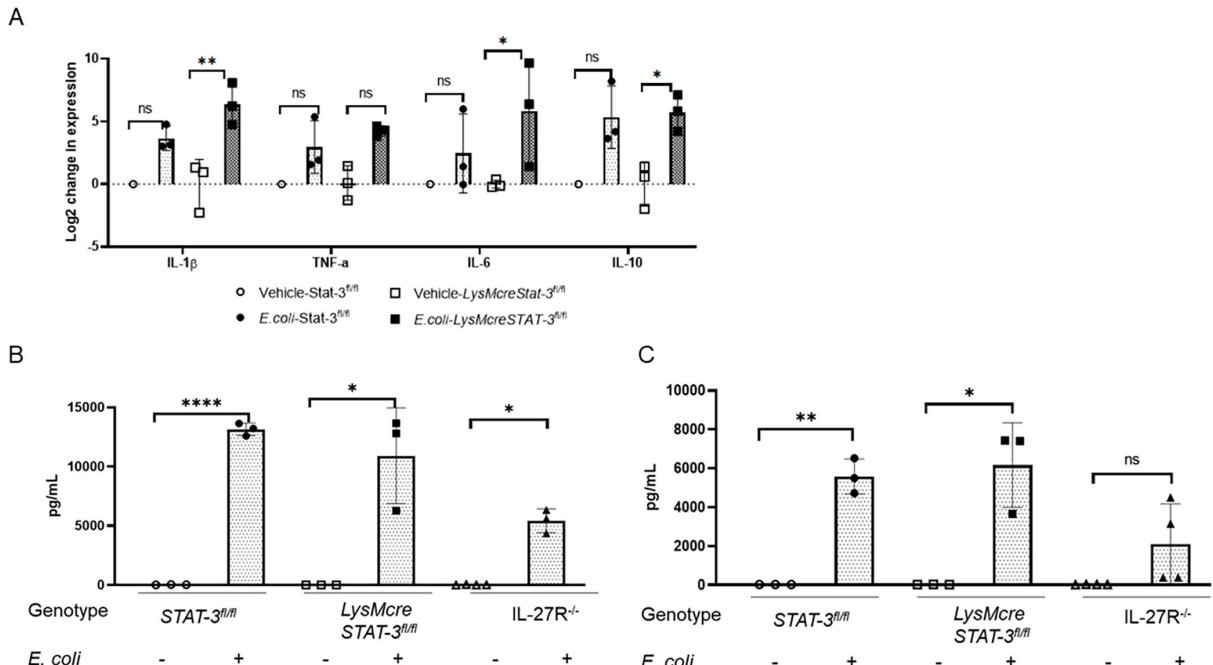

**FIG 4** Inhibition of STAT-3 in myeloid cells promotes increased inflammatory cytokine expression in the spleen during neonatal sepsis. STAT-3$^{fl/fl}$ and LysMcre-STAT-3$^{fl/fl}$ mice were challenged with a target inoculum of 1–3 × 10$^5$ CFU/mouse of *E. coli* O1:K1:H7 or PBS as a control on day 4 of life. (A) Spleens were harvested at 24 h post-infection, and RNA was isolated. The expression of TNF-α, IL-1β, IL-6, and IL-10 was determined in infected spleens relative to uninfected control spleens by real-time PCR using the formula 2-ΔΔCt. (B-C) Blood was collected at 24 h post-infection, and serum levels of (B) IL-6 and (C) IL-10 were measured by multiplex immunoassay. (A-C) Individual animal findings and group means are shown for two combined experiments. Analysis of variance (ANOVA) and unpaired t-tests were used to determine statistical significance in the 95% CI between control and infected groups of different genotypes.

## Global pharmacologic inhibition of STAT-3 promotes increased morbidity in septic neonatal mice

Targeted disruption of STAT-3 in myeloid cells, while effective at reducing responsiveness to IL-27, may impact regulation of the complete immune response in other ways. Indeed, myeloid-specific STAT3 deficiency has been reported to promote exacerbated Th cytokine production, and it is known that IL-27 inhibits the development of regulatory T cells via STAT3 (33, 34). Global genetic disruption of the *STAT-3* gene results in embryonic lethality (35). Considering these ideas and to fully evaluate STAT-3 antagonization as a therapeutic possibility, we also performed studies in neonatal mice during sepsis using S31-201, a pharmacological inhibitor of STAT-3 that blocks DNA binding and transcriptional activity (36). We first validated the inhibition of phosphorylation and translocation of P-STAT-3 in the IL-27-stimulated mouse macrophages pretreated with or without the inhibitor. The potent increase in nuclear translocated P-STAT-3 in response to IL-27 was strongly decreased in the presence of S3I-201 (Fig. S2). Neonatal mice pretreated with S32-201 for 2 h and infected with *E. coli* showed a significant weight reduction and lower blood glucose levels when compared with infected control pups (Fig. 5A and B). Additionally, there was no significant change in bacterial burdens in the blood or spleen observed with S3I-201 treatment compared with infected control pups (Fig. 5C and D). However, in contrast to LysMcreSTAT-3$^{fl/fl}$ neonatal mice, there was no change in mortality in the presence of the inhibitor over 24 h of infection. These findings further support the conclusion that STAT-3 activity does not represent a viable therapeutic target for neonatal sepsis.

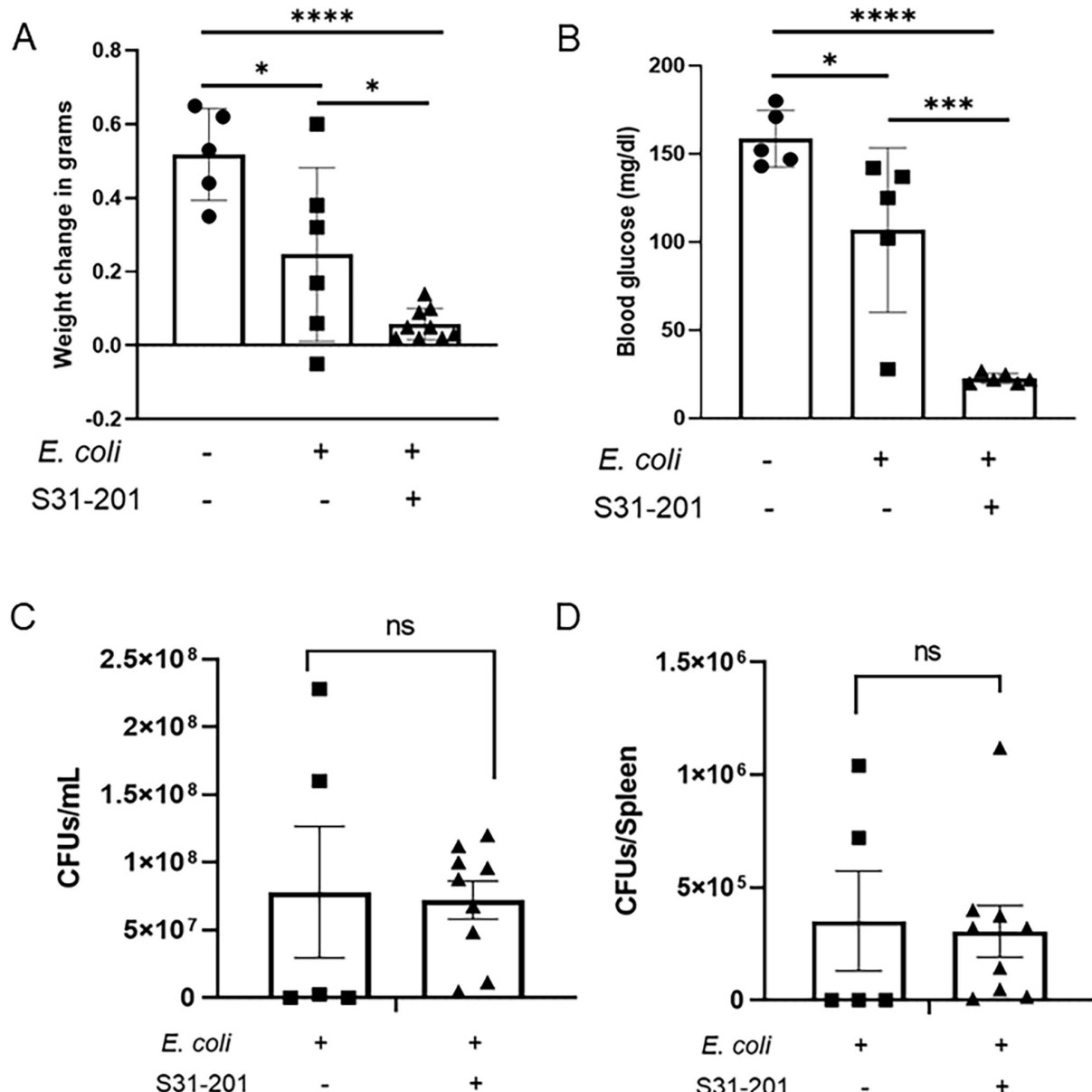

**FIG 5** Pharmacological inhibition of STAT-3 increases morbidity during neonatal sepsis. Neonatal C57BL/6 mice ($n = 3$) were subcutaneously administered S31-201 (5 mg/kg) or PBS ($n = 2$) as a control 2 h prior to infection with *E. coli* O1:K1:H7 at an inoculum range of $1–3 \times 10^5$ CFU/mouse on day 4 of life. At 24 h post-infection, weight change (A) and blood glucose levels (B) were measured. Blood (C) and spleens (D) were collected for bacterial enumeration by standard plate counts. Individual animal findings and experimental group means are shown for three combined experiments. Statistical significance in the 95% CI was determined using a Mann–Whitney test; exact *P*-values are shown.

## DISCUSSION

Neonates are at increased risk for bacterial sepsis, and elevated levels of IL-27 cytokine in early life promote inflammation during neonatal sepsis by directly compromising control of bacteria that drive the inflammatory response (14, 15). In a murine model of neonatal sepsis, mice deficient in IL-27 signaling exhibit reduced mortality, healthy weight gain, and improved control of bacteria with lower systemic inflammation (18). As such, it is important to understand the molecular mechanisms that play a key role in IL-27 regulation of bacterial clearance and related outcomes during neonatal sepsis. Depending on the type and activation state of an immune cell, IL-27 induces the activation of predominantly either STAT-1 or STAT-3 (10). Targeted deletion of STAT-3

in macrophages leads to increased responsiveness to inflammatory stimuli (22–24). In addition, STAT-3 activation in macrophages and neutrophils is essential for anti-inflammatory effects mediated by another immune suppressive cytokine, IL-10 (24). Considering these facts, we studied neonatal sepsis outcomes and clearance of bacterial burdens in the absence of STAT-3 activation to address the role of the STAT-3 signaling pathway in IL-27-dependent suppression of phagocytic activity. Currently, there are no adjunctive therapies to antibiotics that have proven beneficial in the management of neonatal sepsis (37). Several FDA-approved drugs as well as natural compounds have been shown to inhibit STAT-3 activity (38). If targeting STAT-3 could achieve the same outcome as IL-27 interference, these drugs and compounds could be repurposed to provide new therapeutic options for neonatal sepsis.

In this study, we used two complementary strategies to block STAT-3 signaling. Since STAT-3 deficiency is embryonic lethal, we used LysMcreSTAT-3,$^{fl/fl}$ a myeloid-specific STAT-3 deletion model (39). The second method was global inhibition of STAT-3 activation using the pharmacological inhibitor S32-201 (36). Similar to IL-27Rα$^{-/-}$ mice pups, BMDMs isolated from LysMcreSTAT-3$^{fl/fl}$ pups showed improved bacterial clearance relative to WT neonatal BMDMs *in vitro*. Bacterial killing was further and significantly reduced by the addition of IL-27 in WT BMDMs, unchanged in IL-27Rα-deficient BMDMs, and modestly reduced in LysMcreSTAT-3$^{fl/fl}$ BMDMs. The latter may reflect some contributions from other signaling pathways triggered directly or indirectly by IL-27. During phagocytosis, macrophages internalize pathogens and traffic them into phagosomes that fuse with lysosomes to form phagolysosomes. Phagosomal acidification is a critical step in killing the bacteria (40). IL-27 has direct influence on the lysosomal pathway and is known to compromise bacterial killing (41, 42). We observed increased lysosomal acidification in IL-27Rα$^{-/-}$ and LysMcreSTAT-3$^{fl/fl}$ neonatal BMDMs compared with WT. Liu et al. described an association of STAT-3 with cytosolic and lysosomal vacuolar H$^+$-ATPase whereby increased lysosomal acidification is observed upon STAT-3 depletion (43). It seems likely that STAT-3 is part of the IL-27-regulated mechanism in this way. We also observed that infected LysMcreSTAT-3$^{fl/fl}$ neonates were superior at clearing bacteria compared with the control infected pups, but this was more efficient in the spleen (99.5% reduction) than in the blood (68% reduction). Reduced bacterial clearance in the blood may be reflective of the abundance and/or type of myeloid cells in the spleen in which STAT-3 was inactive (44). Whereas the blood is more enriched in monocytes and neutrophils, a greater diversity of macrophage and dendritic cell subtypes exists in the spleen. These cells may be more equipped for bacterial killing in the absence of STAT-3 activation. Indeed, monocytes and neutrophils from STAT-3-deficient patients were not different from healthy controls in fungal killing (45). This contrasts with our findings that focused on bacterial killing in macrophages.

Though improved bacterial clearance was observed, this did not translate to improved signs of morbidity, such as healthy weight gain or maintenance of blood glucose levels, in LysMcreSTAT-3$^{fl/fl}$ pups. Remarkably, we documented mortality of more than half of the infected pups with myeloid deletion of STAT-3. Although we do not expect it is the complete explanation, local heightened inflammation in the tissues may contribute as inflammatory cytokine expression increased significantly in the spleen of LysMcreSTAT-3$^{fl/fl}$ neonates during infection. In contrast, IL-27 receptor KO neonatal pups not only showed improved bacterial clearance but also exhibited reduced levels of inflammatory cytokines in the serum during infection (14). In a separate study, adult mice deficient in hepatic cell STAT-3 activation experienced hyperactive inflammatory responses during sepsis (46). Similarly, STAT-3-deleted cardiac myocytes secreted significantly more TNF-α after LPS treatment and are more susceptible to heart failure than myocytes with intact STAT-3 signaling (47). In line with our findings on control of bacteria, a prior study reported that after infection with *M*. tuberculosis, LysMcreSTAT-3$^{fl/fl}$ mice showed lower bacterial loads in the lungs and spleen (39). However, in this report, lower bacterial numbers were associated with an enhanced ability of STAT-3-deficient antigen-presenting cells to stimulate IL-17 from mycobacterial-specific T cells. A separate

study showed that murine myeloid STAT-3 deficiency did not affect clearance of *S. aureus in vitro* or *in vivo* but did enhance expression of costimulatory molecules as well as matrix metalloproteinase 9 that is implicated in tissue damage (48). Lafdil et al. found that myeloid-specific STAT-3-deficient mice exhibited enhanced activation of STAT-1 and Th1 cell cytokines, IFN-γ and IL-17, that contributed to extensive liver damage (34). This study did not analyze any 24 h post-infection recovery phase parameters, which is one of the limitations of the study. Our results from global inhibition of STAT-3 using a chemical inhibitor did not align with myeloid-specific STAT-3 deletion; there was no improvement of bacterial clearance, and signs of morbidity were significantly worse. However, we did not find increased mortality as observed with myeloid-specific STAT-3-deficient neonates. This approach further validated that inhibition of STAT-3 is not a viable therapeutic consideration for neonatal sepsis.

Taken together, our findings suggest that STAT-3 does contribute to IL-27 suppression of macrophage-mediated bacterial killing; however, neither myeloid deletion nor global inhibition of STAT-3 in neonatal mice showed improvement in outcomes during *E. coli*-induced neonatal sepsis that were observed in IL-27Rα$^{-/-}$ mouse pups. Moreover, our findings also verified that myeloid deletion of STAT-3 in neonates increased lethality during sepsis, and total pharmacological inhibition of STAT-3 could not rescue the neonatal mice. In conclusion, our findings contribute to enhanced understanding of molecular mechanisms that operate during IL-27 regulation of the host response during early life infection. However, they also serve as preclinical studies to inform that while IL-27 remains a viable therapeutic target for neonatal sepsis, antagonization of STAT-3 activity does not invoke the same promise.

## ACKNOWLEDGMENTS

This study was supported by NIH grant AI163333.

## AUTHOR AFFILIATIONS

[1]Department of Microbiology, Immunology, and Cell Biology, West Virginia University School of Medicine, Morgantown, West Virginia, USA
[2]Vaccine Development Center, West Virginia University Health Sciences Center, Morgantown, West Virginia, USA

## AUTHOR ORCIDs

Madhavi Annamanedi  http://orcid.org/0000-0001-5460-6532
Cory M. Robinson  http://orcid.org/0000-0002-2122-2046

## FUNDING

| Funder | Grant(s) | Author(s) |
| --- | --- | --- |
| National Institute of Allergy and Infectious Diseases | AI163333 | Cory M. Robinson |

## AUTHOR CONTRIBUTIONS

Madhavi Annamanedi, Conceptualization, Data curation, Formal analysis, Funding acquisition, Investigation, Methodology, Resources, Supervision, Writing – original draft, Writing – review and editing | Jessica M. Povroznik, Methodology, Writing – review and editing | Cory M. Robinson, Conceptualization, Formal analysis, Funding acquisition, Investigation, Methodology, Resources, Supervision, Writing – original draft, Writing – review and editing

## ADDITIONAL FILES

The following material is available online.

## Supplemental Material

**Supplemental material (Spectrum02211-24-S0001.pptx).** Figure S1 video images and Figure S2.

## Open Peer Review

**PEER REVIEW HISTORY (review-history.pdf).** An accounting of the reviewer comments and feedback.

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
