## [Reviewer comments · Microbiology Spectrum]

Microbiology Spectrum

STAT-3 is necessary for IL-27-mediated macrophage suppression but does not represent a therapeutic target for *E. coli* induced neonatal sepsis

Madhavi Annamanedi, Jessica Povroznik, and Cory Robinson

Corresponding Author(s): Cory Robinson, West Virginia University School of Medicine

Review Timeline:

Submission Date:	September 6, 2024
Editorial Decision:	December 27, 2024
Revision Received:	January 29, 2025
Accepted:	February 27, 2025

Editor: Mariola Ferraro

Reviewer(s): Disclosure of reviewer identity is with reference to reviewer comments included in decision letter(s). The following individuals involved in review of your submission have agreed to reveal their identity: Modher Nagem Abed Abed (Reviewer #4)

Transaction Report:

DOI: <https://doi.org/10.1128/spectrum.02211-24>

Re: Spectrum02211-24 (STAT-3 is necessary for IL-27-mediated macrophage suppression but does not represent a therapeutic target for neonatal sepsis)

Dear Dr. Cory M Robinson:

Thank you for the privilege of reviewing your work. Below you will find my comments, instructions from the Spectrum editorial office, and the reviewer comments.

Revision Guidelines

Sincerely,
Mariola Ferraro
Editor
Microbiology Spectrum

Reviewer #3 (Comments for the Author):

Dear Authors,

Your submitted manuscript is elaborated at a high level. The figures are well presented and clear and easy to understand.

To enhance the clarity of the results, I recommend to add statistical significance markers (e.g., ***, **, or *) besides p values to indicate significant differences between groups in the figures.

Reviewer #4 (Comments for the Author):

No comment

The manuscript submitted by Annamanedi et al., titled “STAT-3 is necessary for IL-27-mediated macrophage suppression but does not represent a therapeutic target for neonatal sepsis” addresses the contribution of IL-27 and STAT-3 signaling axis in bacterial clearance, inflammatory stress, morbidity, and mortality in neonatal mice under *E. coli*-induced *in vitro* and *in vivo* models. The hypothesis that compromised control of bacteria in response to IL-27 during neonatal sepsis requires STAT-3 signaling along its phosphorylation activation state in neonatal sepsis phenotypes (including IL-27Ralpha global deficiency) was experimentally explored. Using bone-marrow-derived macrophages (BMDMs), conditional STAT-3 cre-flox mouse system under LysM-targeted deletion was studied to observe the lack of responsiveness to STAT-3 phosphorylation under IL-27 *in vitro* treatment. Due to improved antimicrobial outcomes recorded in IL-27Ralpha^{-/-} neonatal BMDMs compared to WT, the authors further compared the ability of conditional STAT-3 deficient BMDMs to phagocytose bacteria via lysosomal tracking using fluorescence-based assays. The authors utilize a neonatal mice model under the context that IL-27 is induced at high levels in early days of life and in bacterial sepsis, which further suggests the cytokine signaling relevance as a potential therapeutic target. The authors characterize the morbidity outcomes in neonatal mice based on hypoglycemia, weight loss, and septic bacterial burdens over 24-hours post-induction of *E. coli* challenge. Increased mobility and mortality is reported in the conditional STAT-3 deficient murine neonatal sepsis model compared to STAT-3 control mice, suggesting an intermediate transient effect during infection. The authors also address differential inflammatory responses of IL-27Ralpha^{-/-} and conditional STAT-3 deficient neonate mice, further suggesting some similarities and differences across murine phenotypes. Additionally, global pharmacological use of S3I-201 (or S32-201) as pre-treatment to neonatal sepsis challenge preliminarily indicates no change in bacterial burden nor mortality over 24-h challenge. The authors conclude that results support that IL-27 remains viable to target while STAT-3 induces differential responses that are not beneficial to the neonate during *E. coli* infection.

Major Comments

- 1) The authors apply the use of a sepsis model with one *E. coli* strain as pathogenic organism. Although large proportion of early-onset neonatal sepsis are attributed to this gram-negative organism, generalization of the findings suggests the axis studied in the presented study is not a therapeutic target for the disease seems preliminary. This is due to the fact that polymicrobial sepsis also affects about 15% of all neonatal cases (PMC4226990). Furthermore, line 340-341 correctly reflect the findings from the study, which would suggest to adjust the title to include *E.coli*-induced neonatal sepsis in some manner.
- 2) Line 219-220 states there was no differences in bacterial burden reduction, however, Figure 2A indicates a trending non-significant decrease upon IL-27 treatment in LysMcreSTAT-3fl/fl mice.

- 3) Figure 4A could be more beneficial to interpretation of results if statistical analysis could be conducted for E.coli-STAT3fl/fl vs. E.coli- LysMcreSTAT3fl/fl groups comparisons (black circles vs. black squares).
- 4) Discussion is well referenced and written. However; could authors comment why the pharmacological inhibitor effects are expected to have a difference as a pre-treatment effects vs. STAT-3 effects mediated during later phases of sepsis phase or recovery phase?

Minor Comments

- 1) Line 52-56 needs reference(s).
- 2) Line 89 "ability to activate STAT-3 activation" should be reworded.
- 3) Line 100 in methods section: add primer sequences for genotyping mice.
- 4) Line 101 says pups, 4 day old pups should be added into the sentence for clarity.
- 5) Line 114: the immunofluorescent staining states that anti-rabbit phosphor-STAT-3 antibody was used, would this be a rabbit anti-mouse? Including catalog numbers of these reagents might be important to enhance reproducibility.
- 6) Line 123 "Bacterial clearance by BMDMs in vitro was measured as described previously (Seman 2020)." should be changed to reference number to match style of the manuscript.
- 7) Could the authors comment on any potential baseline differences to unphosphorylated STAT-3 across conditions evaluated in Figure 1A-B and Supplemental Figure 1? Quantification of the images in the figures would be beneficial in some way to improve interpretation of results.
- 8) Figure 2B need more context in the results section to describe differences at baseline across murine genotypes.
- 9) Figure 3 legend letters need to be bold to match the rest of the manuscript legends.
- 10) Line 230-231: include reference if manuscript published
- 11) Line 233-235 and Figure 3C-D show percentage reduction in bacterial burden in blood and spleen. More number of biological replicates and reporting mean CFU/ml or Log(CFU/mL) should be included in the results section.
- 12) Figure 4A-B is missing LysMcreSTAT-3 control group without E. coli challenge.
- 13) Figure 4C needs more biological number of replicates to conduct appropriate comparative analysis to LysMcreSTAT-3fl/fl group.
- 14) Figure 4A seems preliminary due to limited biological replicates, and further clarify whether murine IL-1 or IL-1beta was assessed.
- 15) The study addresses the role during the acute phase of E.coli neonatal acute infection phase, but a current limitation of the study is that effects could be part of the recovery phase (> 24-hpi), post-recovery signs of persistent inflammation and fibrosis of peripheral organs were not assessed the conditions studied.
- 16) Figure 5C-D seems preliminary due to limited infected biological replicates under the analysis performed. More biological replicates and a figure legends are further needed for figure interpretation.

Conclusion: Accept with revisions.

January 27, 2025

Dear Dr. Ferraro:

Thank you for the review of our manuscript (Spectrum02211-24) and the opportunity to submit a revised version as a new manuscript. We have carefully evaluated the concerns and comments and have addressed these below. We are grateful for the careful assessment of our work. Our responses follow reviewer comments in bold print. We have included a revised manuscript marked with changes. We look forward to your reply regarding our revision.

Sincerely,

Cory Robinson

Reviewer #1

Major Comments:

- 1) The authors apply the use of a sepsis model with one *E. coli* strain as pathogenic organism. Although large proportion of early-onset neonatal sepsis are attributed to this gram-negative organism, generalization of the findings suggests the axis studied in the presented study is not a therapeutic target for the disease seems preliminary. This is due to the fact that polymicrobial sepsis also affects about 15% of all neonatal cases (PMC4226990). Furthermore, line 340-341 correctly reflect the findings from the study, which would suggest to adjust the title to include *E. coli*-induced neonatal sepsis in some manner.

We understand the concern and agree that we can only make conclusions based on the specific model used. We have changed the title to “STAT-3 is necessary for IL-27-mediated macrophage suppression but does not represent a therapeutic target for *E. coli*-induced neonatal sepsis”.

- 2) Line 219-220 states there was no differences in bacterial burden reduction, however, Figure 2A indicates a trending non-significant decrease upon IL-27 treatment in LysMcreSTAT-3^{fl/fl} mice.

We agree that there is some change with the addition of IL-27 to the LysMcreSTAT-3^{fl/fl} macrophages and the level of bacterial killing is not equivalent to the without IL-27 treatment control. Unlike the macrophages from IL-27R-deficient mice, those from LysMcreSTAT-3^{fl/fl} can still respond to IL-27, just not through STAT-3 signaling. We suspect that the contributions from other signaling pathways and mechanisms that are still responsive allows for some change, albeit not significant. This could suggest that while STAT-3 mediates suppressive activity by IL-27, there are

contributions from other signaling mechanisms. We have modified the text surrounding this data to reflect the significantly decreased killing with WT neonatal macrophages in response to IL-27.

- 3) Figure 4A could be more beneficial to interpretation of results if statistical analysis could be conducted for E.coli-STAT3fl/fl vs. E.coli- LysMcreSTAT3fl/fl groups comparisons (black circles vs. black squares).

We understand the Reviewer's interest in these comparisons. The baseline expression for each genotype could be different, and as such, we have considered showing the change in gene expression within a genotype to be the fairest representation. We believe normalizing across genotypes is problematic for this reason. Since the increases above uninfected controls are significantly different for IL-1, IL-6, and IL-10 in the LysMcreSTAT-3^{fl/fl} but not STAT-3^{fl/fl} mice, we believe that indicates a higher degree of inflammation.

- 4) Discussion is well referenced and written. However, could authors comment why the pharmacological inhibitor effects are expected to have a difference as a pretreatment effects vs. STAT-3 effects mediated during later phases of sepsis phase or recovery phase?

Thank you for your appreciation. We have established that IL-27 levels at baseline in naïve animals is increased relative to older populations (PMID: 23464355, PMID: 30575117). As such, an associated baseline level of STAT-3 activation is also anticipated. Considering this, we examined inhibitor treatment before and after the infection.

Minor Comments:

Thank you for your time and attention to details in finding minor mistakes in the manuscript. We have included corrections accordingly.

- 1) Line 52-56 needs reference(s).
We apologize for the oversight and have added references.
- 2) Line 89 "ability to activate STAT-3 activation" should be reworded.
We have made this correction.
- 3) Line 100 in methods section: add primer sequences for genotyping mice.
These have been included in the revised manuscript.
- 4) Line 101 says pups, 4 day old pups should be added into the sentence for clarity.
This has been included in the manuscript text as requested.
- 5) Line 114: the immunofluorescent staining states that anti-rabbit phosphor-STAT-3 antibody was used, would this be a rabbit anti-mouse? Including catalog numbers of these reagents might be important to enhance reproducibility.

Thank you for finding this typo. We corrected and included catalog numbers as suggested.

- 6) Line 123 "Bacterial clearance by BMDMs in vitro was measured as described previously (Seman 2020)." should be changed to reference number to match style of the manuscript.

We apologize this was overlooked in the original submission and have added the reference.

- 7) Could the authors comment on any potential baseline differences to unphosphorylated STAT-3 across conditions evaluated in Figure 1A-B and Supplemental Figure 1? Quantification of the images in the figures would be beneficial in some way to improve interpretation of results.

There are non-significant baseline STAT-3 levels observed per cell among study groups. We intended to show nuclear translocation. Quantification for the Figure 1 panels A and B have been included and Supplementary Figure 1 contains live cell imaging videos.

- 8) Figure 2B need more context in the results section to describe differences at baseline across murine genotypes.

This has been added in the revised manuscript text.

- 9) Figure 3 legend letters need to be bold to match the rest of the manuscript legends.

This has been corrected accordingly.

- 10) Line 230-231: include reference if manuscript published.

Since the original submission, this work has been accepted for publication and cited accordingly.

- 11) Line 233-235 and Figure 3C-D show percentage reduction in bacterial burden in blood and spleen. More number of biological replicates and reporting mean CFU/ml or Log(CFU/mL) should be included in the results section.

Mean CFU values were included in the Results section to accompany the percentage changes as requested.

We included 3 pups in each group and performed 3 independent experiments. Because of the gastric issues and pregnancy complications in the homozygous *LysMcreSTAT-3^{fl/fl}* mice, pups of the desired genotype are difficult to breed. These complications with STAT-3 deficiency are documented in other literature (PMIDs: 17233738, 10023769). Heterozygous matings yield small numbers of *LysMcreSTAT-3^{fl/fl}* pups with a maximum of 3 among a litter of 7-8 pups from one dam after conforming by genotyping. Out of 9 pups from 3 experiments (n=3), because of the death of 5 pups in infected *LysMcreSTAT-3^{fl/fl}* genotype only 4 live pups burdens were represented in the figure 3C-D.

- 12) Figure 4A-B is missing *LysMcreSTAT-3* control group without *E. coli* challenge.

The LysMcreSTAT-3 control group without *E. coli* challenge was included and termed as vehicle- LysMcreSTAT-3 in the figure 4A with symbol and 3rd bar in the figure 4B.

13) Figure 4C needs more biological number of replicates to conduct appropriate comparative analysis to LysMcreSTAT-3fl/fl group.

As stated earlier in comment 11, restrictions of the neonatal model and breeding complications of LysMcreSTAT-3^{fl/fl} prevented more than 3 biological replicates per experiment.

14) Figure 4A seems preliminary due to limited biological replicates, and further clarify whether murine IL-1 or IL-1beta was assessed.

We understand the reviewer concern, however, the limitations were described above. Additionally in this case, some of the spleens were committed to measurement of the bacterial burden and others for gene expression within the same experiment to maintain matching infectious doses for both outcomes and to minimize biological variation. Within each genotype and experimental condition, the cytokine expression values from the all the study animals showed a similar trend.

We measured IL-1 beta and this detail has been added in the manuscript.

15) The study addresses the role during the acute phase of *E. coli* neonatal acute infection phase, but a current limitation of the study is that effects could be part of the recovery phase (> 24-hpi), post-recovery signs of persistent inflammation and fibrosis of peripheral organs were not assessed the conditions studied.

Yes, we do agree with the reviewer that we didn't check recovery phase effects and included in the manuscript as one of the limitations of the study.

16) Figure 5C-D seems preliminary due to limited infected biological replicates under the analysis performed. More biological replicates and a figure legend are further needed for figure interpretation.

With the limitation of the neonatal study model, we were able to include data from maximum 3 biological replicates in each experiment. Within an average litter size of 7-8 pups, 2 were committed as uninfected controls, 2 infected, and 3 infected while receiving inhibitor. Additionally, only surviving pups can be evaluated for signs of morbidity and to measure bacterial burdens. These details are updated in the revised figure legend.

Reviewer #3

Your submitted manuscript is elaborated at a high level. The figures are well presented and clear and easy to understand. To enhance the clarity of the results, I recommend to add statistical significance markers (e.g., ***, **, or *) besides p values to indicate significant differences between groups in the figures.

Thank you for your comments and we have marked statistical significance with asterisks as suggested by the reviewer and updated the figures and legends.

Reviewer #4

This reviewer did not have comments in need of address.

Re: Spectrum02211-24R1 (STAT-3 is necessary for IL-27-mediated macrophage suppression but does not represent a therapeutic target for *E. coli* induced neonatal sepsis)

Dear Dr. Cory M Robinson:

Your manuscript has been accepted, and I am forwarding it to the ASM production staff for publication. Your paper will first be checked to make sure all elements meet the technical requirements. ASM staff will contact you if anything needs to be revised before copyediting and production can begin. Otherwise, you will be notified when your proofs are ready to be viewed.

Sincerely,
Mariola Ferraro
Editor
Microbiology Spectrum

Reviewer #3 (Comments for the Author):

The issues were all solved.